# 1,3,5-Triazine Nitrogen Mustards with Different Peptide Group as Innovative Candidates for AChE and BACE1 Inhibitors

**DOI:** 10.3390/molecules26133942

**Published:** 2021-06-28

**Authors:** Dawid Maliszewski, Agnieszka Wróbel, Beata Kolesińska, Justyna Frączyk, Danuta Drozdowska

**Affiliations:** 1Department of Organic Chemistry, Medical University, Mickiewicza Street 2a, 15-222 Białystok, Poland; dawid.maliszewski@umb.edu.pl; 2Institute of Organic Chemistry, Faculty of Chemistry, Lodz University of Technology, Zeromskiego 116, 90-924 Lodz, Poland; beata.kolesinska@p.lodz.pl (B.K.); justyna.fraczyk@p.lodz.pl (J.F.)

**Keywords:** acetylcholinesterase, *β*-secretase, nitrogen mustards, 1,3,5-triazine

## Abstract

A series of new analogs of nitrogen mustards (**4a**–**4****h**) containing the 1,3,5-triazine ring substituted with dipeptide residue were synthesized and evaluated for the inhibition of both acetylcholinesterase (AChE) and *β*-secretase (BACE1) enzymes. The AChE inhibitory activity studies were carried out using Ellman’s colorimetric method, and the BACE1 inhibitory activity studies were carried out using fluorescence resonance energy transfer (FRET). All compounds displayed considerable AChE and BACE1 inhibition. The most active against both AChE and BACE1 enzymes were compounds **A** and **4a**, with an inhibitory concentration of AChE IC_50_ = 0.051 µM; 0.055 µM and BACE1 IC_50_ = 9.00 µM; 11.09 µM, respectively.

## 1. Introduction

Medicinal chemistry plays a crucial role in the design, synthesis, and development of novel bioactive molecules, and this allows for the potential use of different drug discovery strategies. During the last decade, new concepts in drug design and development called either multi-target directed ligand (MTDL), multiple ligand strategy (MLS), or designed multiple ligands (DMLs), have been developed as an innovative approach for complex diseases [1,2,3].

With its three-fold symmetry, 1,3,5-triazine is an important heterocyclic skeleton offering a versatile template that facilitates in the selective modification of the substituents on the ring [4,5]. Owing to the immense and varied bioactivities of triazine derivatives, e.g., as an anticancer [5] antimicrobial [6], antiviral [7], antimalarial [8], antitubulin [9], and anti-inflammatory [7], several attempts have been made to modify the structure of triazine derivatives to improve their activities and to search for new attractive targets [10]. For example, 1,3,5-triazine derivatives were found to possess a selective antagonist effect against adenosine receptors [11] and were also found to be effective against various cysteine cathesins with endopeptidase activity [4]. It has also been reported that modifications at the 2-, 4-, and 6- positions of the triazine ring have allowed a wide group of derivatives associated with anti-cancer, anti-inflammatory, antibacterial properties, among others, to be obtained [12,13,14]. A large group of tested 1,3,5-triazine derivatives contained groups with conditioning alkylating properties, enabling reactions with most nucleophilic functional groups of proteins and nucleic acids [15]. In our previous work, we presented a synthesis of triazine derivatives bearing one, two, or three 2-chloroethylamine residues [16], which can be classified as derivatives of nitrogen mustards (NMs) used as antiproliferative agents in the treatment of ovarian carcinoma, leukemia, lymphoma, and multiple myelomas [17,18]. This classification is based on the presence of the 2-chloroethylamine motif characteristic of nitrogen mustards. A significant finding was that in the case of estrogen-dependent breast cancer MCF-7, which is resistant to chemotherapy, activity increased with the number of 2-chloroethylamino moieties (compound **A**) (Figure 1) [16].

In our additional studies, we found that a series of novel 2,4,6-trisubstituted 1,3,5-triazine derivatives bearing 2-chloroethyl and peptide moieties [19] were also active against cancer cells. The most cytotoxic derivative was triazine with an AlaAla-OMe substituent on the 1,3,5-triazine ring (compound **B**) (Figure 1, panel 1).

On the other hand, triazine derivatives also show a variety of biological activity useful in the search for new therapeutics with potential utility in the treatment of Alzheimer’s disease (AD). AD is a complex progressive neurodegenerative disorder that is characterized by the progressive and chronic deterioration of memory and other cognitive functions [20]. Although the disease is multifactorial and heterogeneous, the predominant neuropathological hallmarks of AD are a massive loss of cholinergic neurons, deposition of neurofibrillary tangles, and amyloid plaques in the brain [21].

The precise mechanism of Aβ deposition is currently not entirely understood, however it is a critical feature and the first initiating pathological event of AD [22]. Various studies have strongly supported that the Aβ theory is a central strategy for developing effective treatments for AD [23,24,25].

Acetylcholinesterase (AChE) and β-secretase (BACE1) are key enzymes in the pathogenesis of AD, and their inhibition is of particular importance in Alzheimer’s disease treatment [26,27]. Numerous studies have demonstrated that AChE accelerates Aβ aggregation and amyloid fibril plaque formation. Furthermore, it influences the conformational and biochemical changes of Aβ, which accelerate Aβ fibrils by forming stable AChE-Aβ complexes [28]. As a result, dual enzymatic inhibition of BACE1 and AChE, as well as inhibition of Aβ aggregation, can be proposed as a promising and preventive route against AD [21].

Jameel et al. developed and synthesized a series of triazine triazolopyrimidines for the treatment of Alzheimer’s disease [29]. 1,3,5-Triazine derivatives **C** and **D** showed the highest potential of AChE inhibition, with IC_50_ values of 0.065 and 0.092 μM, respectively (Figure 1, panel 1). [30].

It has been found that 6-amino-4-phenyl-3,4-dihydro-1,3,5- triazin-2(1H)-ones are the first class of molecules able to simultaneously modulate BACE-1 and GSK-3β [31].

Triazinone **E** showed well-balanced in vitro potencies against the two enzymes (IC_50_ of (18.03 ± 0.01) μM and (14.67 ± 0.78) μM for BACE-1 and GSK-3β, respectively). In cell-based assays, it displayed effective neuroprotective and neurogenic activities and no neurotoxicity [31]. Yazdani et al. evaluated ten new 3-hydrazinyl-1,2,4-triazines bearing pendant aryl phenoxymethyl-1,2,3-triazole as multifunctional ligands against AD. It was shown that compounds containing Cl and NO_2_ groups at the para position of the phenyl ring, namely compounds **F** (IC_50_ = 8.55 ± 3.37 μM) and **G** (IC_50_ = 11.42 ± 2.01 μM) (Figure 1, panel 2), possess promising BACE1 inhibitory potential. Molecular docking analysis of compounds **F** and **G** showed that the high BACE1 inhibitory potential may be in part due to the hydrogen bond interactions with Asp32 and Asp228 within the catalytic cavity of the BACE1 active site [32]. These results demonstrate that 1,2,4-triazine derivatives bearing an aryl phenoxy methyl-1,2,3-triazole have promising properties as therapeutic agents for AD [32].

Iraji et al. synthesized, characterized, and evaluated a series of novel triazine analogs as MTLs for AD. Compounds **H** and **I** (Figure 1, panel 2) were found to possess significant BACE1 inhibitory properties with IC_50_ values of 0.91 (±0.25) μM and 0.69 (±0.20) μM, respectively. Docking evaluations provided insight into enzyme inhibitory interactions of novel synthesized compounds with the BACE1 active site involving the critical role of Gln73 and/or Phe108 alongside Asp32. These findings demonstrate the high potential of triazine scaffolds in the design of MTLs for the treatment of AD [33].

Taking into account the multi-target directed ligand (MTDL) strategy used in medical chemistry, as part of the presented research, we attempted to design, synthesize, and test a series of eight new analogs, (**4a**–**4h**) nitrogen mustard derivatives containing the 1,3,5-triazine ring substituted with dipeptide residue (Figure 2). 

The compounds planned for synthesis contained a 2-chloroethyloamino fragment attached to 1,3,5-triazine via the piperazine ring and dipeptide sequence containing an Ala-OMe fragment substituent on the 1,3,5-triazine ring. Compounds **4a**–**4h**, which were designed in this way, can constitute analogs of compounds **A** and **B**. Additionally, we planned to check in vitro inhibitory activity towards cholinesterases (AChE) and β-secretase (BACE1) for all of the synthesized compounds, which is consistent with the strategy of multi-target drugs. As such, the expectation is that they will show satisfactory activity against both enzymes, and thus will be promising molecules useful in the treatment of Alzheimer’s disease.

## 2. Results and Discussion

### 2.1. Chemistry

When designing the structures of compounds **4a**–**4h**, it was deemed necessary to eliminate the possibility of a reaction of the chloroethylamine group with functional groups in the side chains of amino acids of dipeptides. Therefore, in the synthesized derivatives, the amino, hydroxyl, guanidine, and carboxyl functions of the side chains of amino acids were protected. In addition, it was assumed that in the preliminary studies, it would be justifiable to introduce (i) basic amino acid residues (lysine, arginine, histidine) into the peptide fragment because it was expected that these amino acids would form ionic bonds with carboxylate anions in the active center of enzymes and facilitate the interaction with phospholipid cell membranes, and to also introduce (ii) amino acid residues (derivative **4c**, **4g**, **4h**, and **4d**) with an aromatic ring present in the amino acid or as a fragment of the protective group, in order to determine the influence of π-π or hydrophobic interactions between the substrate and the active site of the enzyme. On the other hand, it was expected that it would be possible to test the influence of aromatic amino acids on the inhibition of β-secretase (BACE1) activity, which is known to hydrolyze beta-site amyloid precursor proteins, and the formation of β-amyloid structures results in the formation of insoluble amyloid deposits. It is known that both β-amyloid and other aggregates of their hot-spots contain aromatic amino acids [34,35,36].

The synthesis of new analogs of nitrogen mustards **4a**–**4h** containing the 1,3,5-triazine ring substituted with dipeptide residue is a multi-step process (Scheme 1). The first stage is the reaction of dipeptides (**1a**–**h**) showing an unprotected amino group with 2,4-dichloro-6-methoxy-1,3,5-triazine (DCMT) (**2**). NaHCO_3_ was used as a hydrogen chloride acceptor solid. Derivatives **3a**–**3h** were obtained with a yield in a range from 82.3% to 99.2%. The 1,3,5-triazine ring includes chloride, thus tertiary amines react, leading to the formation of triazinylammonium chloride as an intermediate. As a tertiary amine, DABCO has the ability of alkylating the forming compounds **4a–h** with the 2-chloroethylamine group, which is characteristic of nitrogen mustards. At the end of this process, eight new nitrogen mustard analogs with a dipeptide fragment attached on the 1,3,5-triazine ring were synthetized (Scheme 1).

As a result of the simulation of the selected parameters (BIOVIA program) characterizing potentially active molecules, it is easy to notice that the only parameter significantly differentiating **A** and **4a**–**4h** derivatives (Table 1) is the polar surface area, which for derivative **A** is lower than 100, and for the remaining **4a**–**4h** compounds, this value is higher than 100. Compound **A** also meets the assumptions of Lipinski’s rule [37], which states that the value of the hydrogen bond acceptors should not exceed 10. For compounds **4a**–**4h**, the determined values are 10 or slightly higher.

Lipinski’s rule of not exceeding five hydrogen bond donors was met for all of the tested compounds. Regarding the postulate that the molecular mass should be less than 500 daltons, for most of the derivatives, the molar mass was slightly higher than the assumed value. On the other hand, an octanol-water partition coefficient (AlogP) parameter exceeded the value of 5, but only for compound **4c**.

### 2.2. In Vitro AChE and BACE1 Inhibitory Activity

The inhibitory activity of triazine derivatives **4a**–**4h** and compound **A** was evaluated in vitro against AChE (from electrophorus electricus) using Ellman’s method and BACE1 (from equine serum) using a Föster resonance energy transfer (FRET) [38]. Donepezil, tacrine, and quercetin were used as reference standards for comparative analysis. The corresponding IC_50_ values of all of the synthesized derivatives are listed in Table 2. Each experiment was performed three times for all of the compounds and the mean half maximal inhibitory concentration of inhibition of AChE and BACE1 enzymatic activity were calculated as IC50 (µM). The obtained results revealed that most of the compounds displayed significant inhibitory activities against both enzymes (AChE and BACE1), with IC_50_ values in the micromolar range (ranging from 0.051 to 1.44 µM and from 9.00 to 58.09 µM, respectively). Moreover, none of the compounds exhibited higher activity than the reference compound donepezil (AChE IC_50_ = 0.046 μM), whereas all compounds, except compounds **4d**, **4e**, and **4f** with IC_50_ values of 0.387; 0.789; 1.44 μM, respectively, were more active than tacrine (AChE IC_50_ = 0.274 μM). Concerning the BACE1 enzyme activity results, compound **A,** with a value of IC_50_ = 9.00 μM, possessed the most similar inhibitory activity to the standard quercetin (IC_50_ = 4.89 μM).

In particular, the most potent compound was **A**, with an inhibitory concentration of AChE IC_50_ = 0.051 µM and BACE1 IC_50_ = 9.00 µM, possibly owing to the presence of a 2-chloroethyloamino fragment. Compounds such as **4a**, **4b,** and **4h**, where the 1,3,5-triazine rings were substituted with Lys-Ala-OMe, Asp-Ala-OMe, and His-Ala-OMe groups, respectively, exhibited significant inhibition of AChE (IC_50_ = 0.055; 0.065; 0.067 µM). Moderate inhibition was seen in **4c**, **4e**, **4f**, and **4g** (IC_50_ = 0.114; 0.387; 0.789; 0.122 µM, respectively), while **4a** and **4h** demonstrated high activity for BACE1 (IC_50_ = 11.09 and 14.25 µM, respectively). It was reported that compounds **4b**, **4c**, and **4g** presented moderate inhibition (IC_50_ = 33.82; 18.09; 28.09 µM, respectively). It was assumed that the Lys-Ala-OMe and His-Ala-OMe groups substituting the 1,3,5-triazine ring analogs have high potential for inhibiting BACE1.

The Alzheimer’s therapy approaches also focused on target self-induced A-β1-42 aggregation, which causes undesirable aggregates and fibrils to form. The structural planarity as well as intercalation ability between β-amyloid sheets renders triazine as a privileged scaffold for designing the multitarget hybrids. The presented compounds could display considerable anti-aggregation potential due to the presence of the triazine scaffold with different active groups, thereby generating the β-amyloid disaggregation effect. The test structures could restrict conformational transition, thereby impeding the formation of a β-sheet structure for further aggregation into mature fibrils [39].

## 3. Materials and Methods

### 3.1. Synthesis

#### 3.1.1. General Information

Thin layer chromatography experiments (TLC) were carried out on silica gel (Merck; 60Å F254). Spots were located with a UV light (254 and 366 nm) and 1% ethanolic 4-(4-nitrobenzyl)pyridine (NBP). Analytical RP-HPLC was performed on a Waters 600S HPLC system (Waters 2489 UV/VIS detector, Waters 616 pump, Waters 717 plus autosampler, HPLC manager software from Chromax) using a Vydac C18 column (25 cm × 4.6 mm, 5 mm; Sigma). HPLC was performed with a gradient of 0.1% TFA in H_2_O (A) and 0.08% TFA in CH_3_CN (B) at a flow rate of 1 mL/min with UV detection at 220 nm, tR in min. MS analysis was performed on an MS Bruker microTOFQIII. Infrared spectra were recorded as KBr pellets or on film using a Bruker ALPHA spectrometer or a PerkinElmer Spectrum 100. 1H-NMR, and 13C-NMR spectra were recorded on a Bruker Avance II Plus (Bruker Corporation, Billerica, MA, USA) spectrometer (700 MHz). Chemical shifts (ppm) were relative to TMS, used as an internal standard. Multiplicities are marked as s = singlet, d = doublet, t = triplet, q = quartet, qu = quintet, m = multiplet.

#### 3.1.2. Synthesis of **3a–h** from DCMT and Dipeptides

##### Synthesis of 2-Chloro-4-Methoxy-6-(NH-Lys(Boc)-Ala-OMe)-1,3,5-Triazine (**3a**). General Procedure

The mixture of solid NaHCO_3_ (0.42 g, 5 mmol) in dichloromethane (DCM) (3 mL) was cooled to 0 °C through vigorous stirring. 2,4-dichloro-6-methoxy-1,3,5-triazine (DCMT) (0.270 g, 1.5 mmol) in DCM (3 mL) was added dropwise to the chilled solution, followed by the slow addition of a solution of H_2_N-Lys(Boc)-Ala-OMe (0.497 g, 1.5 mmol). The reaction continued until the complete consumption of the DCMT. The DCMT stain was monitored with a 0.5% solution of NBP in ethanol at room temperature using TLC analysis. To remove the solvent, after filtration of the solid deposit, a vacuum evaporator was used and the residue was dried under a vacuum with P_2_O_5_ and KOH to a constant weight, giving 0.685 g of product (**3a**), yield = 96.2% as colorless oil.

^1^H NMR (700 MHz, CDCl_3_): δ 1.27 (quint, 2H, J = 4.48 Hz, CH_2_-CH_2_-CH_2_); 1.41 (d, 3H, J = 5.18 Hz, CH_3_-CH-); 1.43 (s, 9H, (CH_3_)_3_-C); 1.57 (quint, 2H, J = 4.28 Hz, -CH_2_-); 1.93 (dt, 2H, J_1_ = 5.25 Hz, J_2_ = 7.52 Hz, -CH-CH_2_-); 3.22 (t, 2H, J = 5.81 Hz, NH-CH_2_-); 3.71 (s, 3H, CH_3_OCO); 4.01 (s, 3H, CH_3_O-); 4.33 (t, 1H, J = 5.52 Hz, NH-CH_2_CH_2_-); 4.36 (q, 1H, J = 5.18 Hz, CH_3_-CH-) [ppm]. ^13^C NMR (176 MHz, CDCl_3_): δ 17.7, 21.4, 28.2, 29.3, 29.4, 40.5, 47.8, 52.3, 53.6, 54.6, 80.5, 162.2, 156.3, 163.4, 171.7, 172.5, 172.9 [ppm].

##### Synthesis of 2-Chloro-4-Methoxy-6-(NH-Asp(OtBu)-Ala-OMe)-1,3,5-Triazine (**3b**)

Starting materials: solid NaHCO_3_ (0.42 g, 5 mmol), DCMT (0.270 g, 1.5 mmol), H_2_N-Asp(OtBu)-Ala-OMe (0.411 g, 1.5 mmol). Product: 0.612 g (**3b**), yield = 97.6%, colorless oil. ^1^H NMR (700 MHz, CDCl_3_): δ 1.37 (s, 9H, (CH_3_)_3_C); 1.41 (d, 3H, J = 5.18 Hz, CH_3_-CH-); 3.01 (d, 3H, J = 6.23 Hz, CH-CH_2_-); 3.72 (s, 3H, CH_3_OCO); 4.01 (s, 3H, CH_3_O-); 4.36 (q, 1H, J = 5.18 Hz, CH_3_-CH-); 5.03 (t, 1H, J = 6.23 Hz, CH-CH_2_-) [ppm]. ^13^C NMR (176 MHz, CDCl_3_): δ 17.7, 28.1, 37.9, 47.8, 52.3, 53.6, 54.6, 81.4, 162.2, 163.4, 171.7, 172.3, 172.5, 172.9 [ppm].

##### Synthesis of 2-Chloro-4-Methoxy-6-(NH-Trp(Boc)-Ala-OMe)-1,3,5-Triazine (**3c**)

Starting materials: solid NaHCO_3_ (0.42 g, 5 mmol), DCMT (0.270 g, 1.5 mmol), H_2_N-Trp(Boc)-Ala-OMe (0.584 g, 1.5 mmol). Product: 0.784 g (**3c**), yield = 98.1%, colorless oil. ^1^H NMR (700 MHz, CDCl_3_): δ 1.41 (d, 3H, J = 5.18 Hz, CH_3_-CH-); 1.42 (s, 9H, (CH_3_)_3_C); 3.05 (d, 2H, J = 6.82 Hz, CH-CH_2_-); 3.72 (s, 3H, CH_3_OCO); 4.02 (s, 3H, CH_3_O-); 4.36 (q, 1H, J = 5.18 Hz, CH_3_-CH-); 4.80 (t, 1H, J = 6.82 Hz, CH-CH_2_-); 6.98–7.62 (m, 5H, Ar) [ppm]. ^13^C NMR (176 MHz, CDCl_3_): δ 17.8, 27.6, 28.2, 47.8, 52.3, 53.6, 54.6, 81.4, 115.2, 117.1, 118.8, 120.2, 122.9, 123.7, 129.8, 135.3, 149.2, 162.2, 163.4, 171.7, 172.5, 172.9 [ppm].

##### Synthesis of 2-Chloro-4-Methoxy-6-(NH-Ser(Bn)-Ala-OMe)-1,3,5-Triazine (**3d**)

Starting materials: solid NaHCO_3_ (0.42 g, 5 mmol), DCMT (0.270 g, 1.5 mmol), H_2_N-Ser(Bn)-Ala-OMe (0.420 g, 1.5 mmol). Product: 0.615 g (**3d**), yield = 98.1%, colorless oil. ^1^H NMR (700 MHz, CDCl_3_): δ 1.41 (d, 3H, J = 5.18 Hz, CH_3_-CH-); 3.72 (s, 3H, CH_3_OCO); 3.84 (d, 2H, J = 6.78 Hz, CH-CH_2_-); 4.01 (s, 3H, CH_3_O-); 4.36 (q, 1H, J = 5.18 Hz, CH_3_-CH-); 4.481 (t, 1H, J = 6.78 Hz, CH-CH_2_-); 4.55 (s, 2H, CH_2_-O-Ar); 7.37–7.44 (m, 5H, Ar) [ppm]. ^13^C NMR (176 MHz, CDCl_3_): δ 17.7, 38.3, 47.8, 52.3, 54.6, 69.3, 72.8, 127.9, 128.5, 128.9, 138.0, 162.2, 163.4, 171.3, 171.7, 172.9 [ppm].

##### Synthesis of 2-Chloro-4-Methoxy-6-(NH-Aib-Ala-OMe)-1,3,5-Triazine (**3e**)

Starting materials: solid NaHCO_3_ (0.42 g, 5 mmol), DCMT (0.270 g, 1.5 mmol), H_2_N-Aib-Ala-OMe (0.282 g, 1.5 mmol). Product: 0.477 g (**3e**), yield = 95.8%, colorless oil. ^1^H NMR (700 MHz, CDCl_3_): δ 1.36 (s, 6H, (CH_3_)_2_C-); 1.41 (d, 3H, J = 5.18 Hz, CH_3_-CH-); 3.72 (s, 3H, CH_3_OCO); 4.01 (s, 3H, CH_3_O-); 4.36 (q, 1H, J = 5.18 Hz, CH_3_-CH-) [ppm]. ^13^C NMR (176 MHz, CDCl_3_): δ 17.8, 27.1, 48.1, 54.6, 52.3, 58.1, 163.4. 167.9, 171.7, 172.9, 175.6 [ppm].

##### Synthesis of 2-Chloro-4-Methoxy-6-(NH-Arg(NO_2_)-Ala-OMe)-1,3,5-Triazine (**3f**)

Starting materials: solid NaHCO_3_ (0.42 g, 5 mmol), DCMT (0.270 g, 1.5 mmol), H_2_N-Arg(NO_2_)-Ala-OMe (0.456 g, 1.5 mmol). Product: 0.650 g (**3f**), yield = 96.7%, colorless oil. ^1^H NMR (700 MHz, CDCl_3_): δ 1.41 (d, 3H, J = 5.18 Hz, CH_3_-CH-); 1.59 (quint, 2H, J = 4.72 Hz, CH_2_-CH_2_); 1.94 (q, 2H, J = 6.47 Hz, -CH-CH_2_); 3.12 (t, 2H, J = 6.25 Hz, CH_2_-CH_2_); 3.72 (s, 3H, CH_3_OCO); 4.01 (s, 3H, CH_3_O-); 4.33 (t, 1H, J = 6.55 Hz, CH-CH_2_); 4.36 (q, 1H, J = 5.18 Hz, CH_3_-CH-) [ppm]. ^13^C NMR (176 MHz, CDCl_3_): δ 17.7, 25.7, 30.7, 41.0, 47.8, 52.3, 53.6, 54.6, 154.7, 162.2, 163.4, 171.7, 172.5, 172.9 [ppm].

##### Synthesis of 2-Chloro-4-Methoxy-6-(NH-Trp-Ala-OMe)-1,3,5-Triazine (**3g**)

Starting materials: solid NaHCO_3_ (0.42 g, 5 mmol), DCMT (0.270 g, 1.5 mmol), H_2_N-Trp-Ala-OMe (0.434 g, 1.5 mmol). Product: 0.633 g (**3g**), yield = 97.5%, colorless oil.

^1^H NMR (700 MHz, CDCl_3_): δ 1.41 (d, 3H, J = 5.18 Hz, CH_3_-CH-); 3.05 (d, 2H, J = 6.82 Hz, CH-CH_2_); 3.72 (s, 3H, CH_3_OCO); 4.02 (s, 3H, CH_3_O-); 4.36 (q, 1H, J = 5.18 Hz, CH_3_-CH-); 4.80 (t, 1H, J = 6.82 Hz, CH-CH_2_); 6.98–7.62 (m, 5H, Ar) [ppm]. ^13^C NMR (176 MHz, CDCl_3_): δ 17.8, 27.7, 47.8, 52.3, 53.6, 54.6, 109.9, 111.6, 118.7, 120.2, 122.3, 124.4, 127.7, 136.4, 162.2, 163.4, 171.7, 172.5, 172.9 [ppm].

##### Synthesis of 2-Chloro-4-Methoxy-6-(NH-His(Ts)-Ala-OMe)-1,3,5-Triazine (**3h**)

Starting materials: solid NaHCO_3_ (0.42 g, 5 mmol), DCMT (0.270 g, 1.5 mmol), H_2_N-His(Ts)-Ala-OMe (0.592 g, 1.5 mmol). Product: 0.797 g (**3h**), yield = 98.8%, colorless oil. ^1^H NMR (700 MHz, CDCl_3_): δ 1.41 (d, 3H, J = 5.18 Hz, CH_3_-CH-); 2.47 (s, 3H, CH_3_); 3.45 (d, 2H, J = 6.38 Hz, CH-CH_2_); 3.71 (s, 3H, CH_3_OCO); 4.02 (s, 3H, CH_3_O-); 4.36 (q, 1H, J = 5.18 Hz, CH_3_-CH-); 5.03 (t, 1H, J = 6.38 Hz, CH-CH_2_); 7.28–7.79 (m, 6H, Ar) [ppm]. ^13^C NMR (176 MHz, CDCl_3_): δ 17.8, 21.3, 30.5, 47.8, 52.3, 127.5, 129.7, 128.8, 135.1, 162.2, 163.4, 171.7, 172.5, 172.9 [ppm].

#### 3.1.3. Synthesis o2-[4-(2-Chloroethyl)Piperazin-1-yl]-4-Methoxy-6-(Dipeptidyl)-1,3,5-Triazines (**4a–h**)

##### Synthesis of 2-[4-(2-Chloroethyl)Piperazin-1-yl]-4-Methoxy-6-(NH-Lys(Boc)-Ala -OMe)-1,3,5-Triazine (**4a**). General Procedure

To a vigorously stirred solution of 2-chloro-4-methoxy-(NH-Lys(Boc)-Ala-OMe)-1,3,5- triazine (**3a**) (0.475 g, 1 mmol) in DCM (10 mL) cooled to 0 °C in an ice-water bath, 1,4-diazabicyclo[2.2.2]octane (DABCO) (0.112 g, 1 mmol) was added. The amount of **3a** was monitored by TLC analysis staining with 0.5% solution of NBP in ethanol. After complete consumption of **3a**, the sample was removed from the ice-water bath, and the reaction was conducted at room temperature until salt (R_f_ = 0.00) converted into **4a** (R_f_ = 0.45). DCM was removed through evaporation. The residue was dried under a vacuum with P2O5 and KOH to a constant weight, giving 0.580 g of product (**4a**), yield = 98.8% as colorless oil. ^1^H NMR (700 MHz, CDCl_3_): δ 1.27 (quint, 2H, J = 4.48 Hz, CH_2_-CH_2_-CH_2_); 1.41 (d, 3H, J = 5.18 Hz, CH_3_-CH-); 1.43 (s, 9H, (CH_3_)_3_-C); 1.57 (quint, 2H, J = 4.28 Hz, -CH_2_-); 1.93 (dt, 2H, J_1_ = 5.25 Hz, J_2_ = 7.52 Hz, -CH-CH_2_-); 2.66 (t, 2 × 2H, J = 6.32 Hz, N-CH_2_-CH_2_-N); 2.85 (t, 2H, J = 6.24 Hz, N-CH_2_-CH_2_); 3.22 (t, 2H, J = 5.81 Hz, NH-CH_2_); 3.58 (t, 2 × 2H, J = 6.31 Hz, N-CH_2_-CH_2_-N); 3.86 (t, 2H, J = 6.24 Hz, Cl-CH_2_); 3.71 (s, 3H, CH_3_OCO); 4.01 (s, 3H, CH_3_O-); 4.33 (t, 1H, J = 5.52 Hz, NH-CH_2_CH_2_); 4.36 (q, 1H, J = 5.18 Hz, CH_3_-CH-) [ppm]. ^13^C NMR (176 MHz, CDCl_3_): δ 17.7, 21.4, 28.2, 29.3, 29.4, 40.5, 41.2, 44.0, 47.8, 51.8, 52.3, 53.0, 53.6, 54.6, 80.5, 162.2, 156.3, 163.4, 171.7, 172.5, 172.9 [ppm]. HRMS: 588.2994, ([M+H]^+^, C_25_H_44_ClN_8_O_6_^+^; calc. 588.11192).

##### Synthesis of 2-[4-(2-Chloroethyl)Piperazin-1-yl]-4-Methoxy-6-(NH-Asp(OtBu)- Ala-OMe)-1,3,5-Triazine (**4b**)

Starting materials: 2-chloro-4-methoxy-(NH-Asp(OtBu)-Ala-OMe)-1,3,5-triazine (**3b**) (0.418 g, 1 mmol), DABCO (0.112 g, 1 mmol). Product: 0.525 g (**4b**), yield = 99.1%, colorless oil. ^1^H NMR (700 MHz, CDCl_3_): δ 1.37 (s, 9H, (CH_3_)_3_C); 1.41 (d, 3H, J = 5.18 Hz, CH_3_-CH-); 2.66 (t, 2 × 2H, J = 6.32 Hz, N-CH_2_-CH_2_-N); 2.85 (t, 2H, J = 6.24 Hz, N-CH_2_-CH_2_); 3.01 (d, 3H, J = 6.23 Hz, CH-CH_2_); 3.58 (t, 2 × 2H, J = 6.31 Hz, N-CH_2_-CH_2_-N); 3.72 (s, 3H, CH_3_OCO); 3.88 (t, 2H, J = 6.24 Hz, Cl-CH_2_); 4.01 (s, 3H, CH_3_O-); 4.36 (q, 1H, J = 5.18 Hz, CH_3_-CH-); 5.03 (t, 1H, J = 6.23 Hz, CH-CH_2_) [ppm]. ^13^C NMR (176 MHz, CDCl_3_): δ 17.7, 28.1, 37.9, 41.2, 44.0, 47.8, 51.8, 52.3, 53.1, 53.6, 54.6, 81.4, 162.2, 163.4, 171.7, 172.3, 172.5, 172.9 [ppm]. HRMS: 531.2773, ([M+H]^+^, C_22_H_37_ClN_7_O_6_^+^; calc. 531.0175).

##### Synthesis of 2-[4-(2-Chloroethyl)Piperazin-1-yl]-4-Methoxy-6-(NH-Trp(Boc)-Ala-OMe)-1,3,5-Triazine (**4c**)

Starting materials: 2-chloro-4-methoxy-(NH-Trp(Boc)-Ala-OMe)-1,3,5-triazine (**3c**) (0.533 g, 1 mmol), DABCO (0.112 g, 1 mmol). Product: 0.638 g (**4c**), yield = 98.9%, colorless oil. ^1^H NMR (700 MHz, CDCl_3_): δ 1.41 (d, 3H, J = 5.18 Hz, CH_3_-CH-); 1.42 (s, 9H, (CH_3_)_3_C); 2.66 (t, 2 × 2H, J = 6.32 Hz, N-CH_2_-CH_2_-N); 2.85 (t, 2H, J = 6.24 Hz, N-CH_2_-CH_2_); 3.58 (t, 2 × 2H, J = 6.31 Hz, N-CH_2_-CH_2_-N); 3.05 (d, 2H, J = 6.82 Hz, CH-CH_2_); 3.72 (s, 3H, CH_3_OCO); 3.88 (t, 2H, J = 6.24 Hz, Cl-CH_2_); 4.02 (s, 3H, CH_3_O-); 4.36 (q, 1H, J = 5.18 Hz, CH_3_-CH-); 4.80 (t, 1H, J = 6.82 Hz, CH-CH_2_); 6.98–7.62 (m, 5H, Ar) [ppm]. ^13^C NMR (176 MHz, CDCl_3_): δ 17.8, 27.6, 28.2, 41.2, 44.1, 47.8, 51.8, 52.3, 53.0, 53.6, 54.6, 81.4, 115.2, 117.1, 118.8, 120.2, 122.9, 123.7, 129.8, 135.3, 149.2, 162.2, 163.4, 171.7, 172.5, 172.9 [ppm]. HRMS: 646.2837, ([M+H]^+^, C_30_H_42_ClN_8_O_6_^+^; calc. 646.1495).

##### Synthesis of 2-[4-(2-Chloroethyl)Piperazin-1-yl]-4-Methoxy-6- (NH-Ser (Bn)-Ala -OMe) -1,3,5-Triazine (**4d**)

Starting materials: 2-chloro-4-methoxy-(NH-Ser(Bn)-Ala-OMe)-1,3,5-triazine (**3d**) (0.424 g, 1 mmol), DABCO (0.112 g, 1 mmol). Product: 0.531 g (**4d**), yield = 99.0%, colorless oil. ^1^H NMR (700 MHz, CDCl_3_): δ 1.41 (d, 3H, J = 5.18 Hz, CH_3_-CH-); 2.66 (t, 2 × 2H, J = 6.32 Hz, N-CH_2_-CH_2_-N); 2.85 (t, 2H, J = 6.24 Hz, N-CH_2_-CH_2_); 3.58 (t, 2 × 2H, J = 6.31 Hz, N-CH_2_-CH_2_-N); 3.72 (s, 3H, CH_3_OCO); 3.84 (d, 2H, J = 6.78 Hz, CH-CH_2_); 3.86 (t, 2H, J = 6.24 Hz, Cl-CH_2_); 4.01 (s, 3H, CH_3_O-); 4.36 (q, 1H, J = 5.18 Hz, CH_3_-CH-); 4.481 (t, 1H, J = 6.78 Hz, CH-CH_2_); 4.55 (s, 2H, CH_2_-O-Ar); 7.37–7.44 (m, 5H, Ar) [ppm]. ^13^C NMR (176 MHz, CDCl_3_): δ 17.7, 38.3, 41.2, 44.1, 47.8, 51.8, 52.3, 53.0, 54.6, 69.3, 72.8, 127.9, 128.5, 128.9, 138.0, 162.2, 163.4, 171.3, 171.7, 172.9 [ppm]. HRMS: 537.3677, ([M+H]^+^, C_24_H_35_ClN_7_O_5_^+^; calc. 537.0237). 

##### Synthesis of 2-[4-(2-Chloroethyl)Piperazin-1-yl]-4-Methoxy-6-(NH-Aib-Ala-OMe) -1,3,5 -Triazine (**4e**)

Starting materials: 2-chloro-4-methoxy-(NH-AIB-Ala-OMe)-1,3,5-triazine (**3e**) (0.332 g, 1 mmol), DABCO (0.112 g, 1 mmol). Product: 0.436 g (**4e**), yield = 98.3%, colorless oil.

^1^H NMR (700 MHz, CDCl_3_): δ 1.36 s, 6H, (CH_3_)_2_); 1.41 (d, 3H, J = 5.18 Hz, CH_3_-CH-); 2.66 (t, 2 × 2H, J = 6.32 Hz, N-CH_2_-CH_2_-N); 2.85 (t, 2H, J = 6.24 Hz, N-CH_2_-CH_2_); 3.58 (t, 2 × 2H, J = 6.31 Hz, N-CH_2_-CH_2_-N); 3.72 (s, 3H, CH_3_OCO); 3.86 (t, 2H, J = 6.24 Hz, Cl-CH_2_); 4.01 (s, 3H, CH_3_O-); 4.36 (q, 1H, J = 5.18 Hz, CH_3_-CH-) [ppm]. ^13^C NMR (176 MHz, CDCl_3_): δ 17.8, 27.1, 41.2, 44.0, 48.1, 54.6, 51.8, 52.3, 53.0, 58.1, 163.4. 167.9, 171.7, 172.9, 175.6 [ppm]. HRMS: 445.0345, ([M+H]^+^, C_18_H_31_ClN_7_O_4_^+^; calc. 444.9283). 

##### Synthesis of 2-[4-(2-Chloroethyl)Piperazin-1-yl]-4-Methoxy-6-(NH-Arg(NO_2_)-Ala- OMe) -1,3,5-Triazine (**4f**)

Starting materials: 2-chloro-4-methoxy-(NH-Arg(NO_2_)-Ala-OMe)-1,3,5-triazine (**3f**) (0.448 g, 1 mmol), DABCO (0.112 g, 1 mmol). Product: 0.551 g (**4f**), yield = 98.4%, colorless oil.

^1^H NMR (700 MHz, CDCl_3_): δ 1.36 s, 6H, (CH_3_)_2_); 1.41 (d, 3H, J = 5.18 Hz, CH_3_-CH-); 2.66 (t, 2 × 2H, J = 6.32 Hz, N-CH_2_-CH_2_-N); 2.85 (t, 2H, J = 6.24 Hz, N-CH_2_-CH_2_); 3.58 (t, 2 × 2H, J = 6.31 Hz, N-CH_2_-CH_2_-N); 3.72 (s, 3H, CH_3_OCO); 3.86 (t, 2H, J = 6.24 Hz, Cl-CH_2_); 4.01 (s, 3H, CH_3_O-); 4.36 (q, 1H, J = 5.18 Hz, CH_3_-CH-) [ppm]. ^13^C NMR (176 MHz, CDCl_3_): δ 17.8, 27.1, 41.2, 44.0, 48.1, 54.6, 51.8, 52.3, 53.0, 58.1, 163.4. 167.9, 171.7, 172.9, 175.6 [ppm]. HRMS: 445.0345, ([M+H]^+^, C_18_H_31_ClN_7_O_4_^+^; calc. 444.9283).

##### Synthesis of 2-[4-(2-Chloroethyl)Piperazin-1-yl]-4-Methoxy-6-(NH-Trp-Ala-OMe)- 1,3,5- Triazine (**4g**)

Starting materials: 2-chloro-4-methoxy-(NH-Trp-Ala-OMe)-1,3,5-triazine (**3g**) (0.433 g, 1 mmol), DABCO (0.112 g, 1 mmol). Product: 0.541 g (**4g**), yield = 99.2%, colorless oil.

^1^H NMR (700 MHz, CDCl_3_): δ 1.41 (d, 3H, J = 5.18 Hz, CH_3_-CH-); 2.66 (t, 2 × 2H, J = 6.32 Hz, N-CH_2_-CH_2_-N); 3.05 (d, 2H, J = 6.82 Hz, CH-CH_2_); 3.72 (s, 3H, CH_3_OCO); 2.85 (t, 2H, J = 6.24 Hz, N-CH_2_-CH_2_); 3.58 (t, 2 × 2H, J = 6.31 Hz, N-CH_2_-CH_2_-N); 3.86 (t, 2H, J = 6.24 Hz, Cl-CH_2_); 4.02 (s, 3H, CH_3_O-); 4.36 (q, 1H, J = 5.18 Hz, CH_3_-CH-); 4.80 (t, 1H, J = 6.82 Hz, CH-CH_2_); 6.98–7.62 (m, 5H, Ar) [ppm]. ^13^C NMR (176 MHz, CDCl_3_): δ 17.8, 27.7, 41.2, 44.1, 47.8, 51.8, 52.3, 53.1, 53.6, 54.6, 109.9, 111.6, 118.7, 120.2, 122.3, 124.4, 127.7, 136.4, 162.2, 163.4, 171.7, 172.5, 172.9 [ppm]. HRMS: 546.2377, ([M+H]^+^, C_25_H_34_ClN_8_O_4_^+^; calc. 546.0337).

##### Synthesis of 2-[4-(2-Chloroethyl)Piperazin-1-yl]-4-Methoxy-6-(NH-His(Ts)-Ala-OMe)-1,3,5-Triazine (**4h**)

Starting materials: 2-chloro-4-methoxy-(NH-His(Ts)-Ala-OMe)-1,3,5-triazine (3h) (0.538 g, 1 mmol), DABCO (0.112 g, 1 mmol). Product: 0.642 g (**4h**), yield = 98.7%, colorless oil.

^1^H NMR (700 MHz, CDCl_3_): δ 1.41 (d, 3H, J = 5.18 Hz, CH_3_-CH-); 2.47 (s, 3H, CH_3_); 2.67 (t, 2 × 2H, J = 6.32 Hz, N-CH_2_-CH_2_-N); 2.85 (t, 2H, J = 6.24 Hz, N-CH_2_-CH_2_); 3.45 (d, 2H, J = 6.38 Hz, CH-CH_2_); 3.58 (t, 2 × 2H, J = 6.31 Hz, N-CH_2_-CH_2_-N); 3.71 (s, 3H, CH_3_OCO); 3.86 (t, 2H, J = 6.24 Hz, Cl-CH_2_); 4.02 (s, 3H, CH_3_O-); 4.36 (q, 1H, J = 5.18 Hz, CH_3_-CH-); 5.03 (t, 1H, J = 6.38 Hz, CH-CH_2_); 7.28–7.79 (m, 6H, Ar) [ppm]. ^13^C NMR (176 MHz, CDCl_3_): δ 17.8, 21.3, 30.5, 41.2, 44.0, 47.8, 51.8, 52.3, 53.0, 127.5, 129.7, 128.8, 135.1, 162.2, 163.4, 171.7, 172.5, 172.9 [ppm]. HRMS: 651.5463, ([M+H]+, C_27_H_37_ClN_9_O_6_S^+^; calc. 651.1494).

### 3.2. Biological Activity

#### 3.2.1. In Vitro Inhibition Studies on AChE

The inhibitory activity of the target compounds on AChE was assessed using the spectroscopic method of Ellman et al. [38]. The Acetylcholinesterase Inhibitor Screening Kit (catalog number MAK324), Purified AChE (catalog number C3389), and donepezil were purchased from Sigma-Aldrich. Enzyme solutions were prepared by dissolving lyophilized powder in double-distilled water. The compounds were dissolved in DMSO and diluted using 0.1 M KH_2_PO_4_/K_2_HPO_4_ phosphate buffer (pH 7.5) at room temperature to yield the corresponding test concentrations of 1–100 mM. Measurements were done using a clear 96-well flat-bottom plate. The absorbance was read on an Infinite M200 fluorescence spectrophotometer (TECAN, Männedorf, Switzerland) (ex. 412 nm) in duplicate experiments with two control wells: a standard (no enzyme) well and one well containing AChE reference enzyme (no inhibitor control). The experimental procedures for AChE activity assays were performed according to the technical bulletins of the acetylcholinesterase activity assay kit (MAK324; Sigma-Aldrich). Purified AChE was prepared to a concentration of 400 Units/L. A reaction mix for each well of reaction was prepared by mixing into a clean tube: 154 μL of Assay Buffer (catalog number MAK324A) 1 μL of Substrate (100 mM, catalog number MAK324B) and 0.5 μL of DNTB (catalog number MAK324C). The reaction was initiated by the addition of 45 μL assay buffer, 5 μL of the enzyme, and the investigated compounds I-IX (5 μL) to the wells to obtain the final concentrations of 1, 10, 20, 50, and 100 mM. A positive control of donepezil was used in the same range of concentrations. The plate was incubated for 15 min. The reaction mix (150 μL) was then added to each sample, the control (no enzyme), and the no inhibitor control wells. The plate was tapped to mix. Absorbance was measured at 412 nm at 0 min and at 10 min. Acetylcholinesterase activity was calculated as the % of inhibition. Results were calculated as IC_50_ (µM) in Table 2. All samples were assayed in triplicate.

#### 3.2.2. In Vitro Inhibition Studies on β-Secretase (BACE1)

The β-Secretase (BACE1) activity detection kit was purchased from Sigma-Aldrich (catalog number CS0010) and the assay was done according to the technical bulletins for the β-secretase activity assay kit [40]. The assay is based on a convenient method of fluorescence resonance energy transfer (FRET), in which florescence signal enhancement is observed after the substrate is cleaved by BACE1. Measurements were done using a 96-well flat-bottom plate for florescence assay. Stock solutions of all derivatives were prepared in DMSO. Each sample was further diluted in an assay buffer to prepare the appropriate concentrations of the test compounds I-IX (1, 10, 20, 50, and 100 Mm). A total of 20 µL of BACE1 substrate was added to 1–10 µL of each test compound in separate wells of a black 96-well microplate and mixed using gentle pipetting. An amount of 10 µL of BACE1 enzyme solution (diluted 10-fold with fluorescent assay buffer (catalog number F8303) to ~0.3 unit/µL) was added just before reading. Fluorescence was measured at “time zero,” immediately after adding the enzyme and after incubation (37 °C; 2 h). Finally, the fluorescence was read on an Infinite M200 fluorescence spectrophotometer (TECAN, Männedorf, Switzerland) (ex. 320 nm; em. 405 nm) in triplicate experiments with a negative control (no enzyme) and a positive control (supplied enzyme activity). BACE1 activity was calculated as a % of inhibition. Results were calculated as IC_50_ (µM) in Table 2.

## 4. Conclusions

Alzheimer’s disease (AD) is a chronic neurological disease related to a decline in brain and nervous system functions. At present, inhibition of acetylcholinesterase (AChE) to enhance neurotransmitters and retardation of β -amyloid formation through β-secretase (BACE1) inhibition are the two main hypotheses for AD prevention and treatment. Due to the multifactorial etiology of AD, the multitarget-directed ligands (MTDLs) approach is promising in the search for new drugs for AD [41]. According to *Benek’s* team, because MTDs select different targets within the disease pathophysiology to achieve therapeutic synergy, they are therefore considered beneficial in the treatment of AD. Therapeutics targeting only one of the molecular targets associated with AD have not yet been successful in the search for a disease-modifying therapy. Therefore, high hopes are placed on the design and synthesis of potential multi-target drugs (MTDs) that act simultaneously in more than one way. Our experiments, presented here, are part of this current research.

Hybrid compounds based on nitrogen mustard 1,3,5-triazine have great therapeutic potential, as confirmed by numerous reports, including our research. In this paper, we reported the synthesis of a new series of 1,3,5-triazine derivatives substituted with a dipeptide residue, alanine connected with another amino acid. Previously obtained compound **A**, triazine with two 2-chloroethylamino fragments, was also tested for its ability to inhibit enzymes involved in the development of AD. All of the tested compounds were found to be active, both against AChE and BACE1. The most active were derivative **4a**, containing the fragment Lys-Ala, and compound **A**, with the Ala-Ala fragment. Their inhibitory effects against AChE were similar to donepezil. These compounds were also the most effective against BCE1, which may indicate an important role for the presence of alanine in the molecule. 

Further investigations are needed to ascertain whether the studied compounds should be considered for possible therapeutic applications. Molecular docking as well as molecular dynamics calculations are required to explain the mode of action of the new compounds on the indicated molecular targets, which is the next planned stage of our research. Our preliminary studies confirm the value of triazine derivatives to generate hybrid molecules that have multiple beneficial biological activities as potential Alzheimer’s disease modifiers. In-depth research work in this area will be continued.

## Data Availability

Not applicable.

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
