# Peer review of "1,3,5-Triazine Nitrogen Mustards with Different Peptide Group as Innovative Candidates for AChE and BACE1 Inhibitors"

_molecules, 2021, doi:10.3390/molecules26133942_

Round 1
Reviewer 1 Report
GENERAL COMMENTS
In this MS, a series of 8 compounds, containing the 1,3,5-triazine core substituted with dipeptide residues and a nitrogen mustard moiety, were synthesized and assayed in terms of inhibition of both AChE and BACE1. The authors continued herein their work on the development of triazine compounds (refs. 33 and 34) and triazine-mustard compounds (ref. 34)
Overall, the work described in this MS has a significant weight on organic synthesis, it is quite interesting, but the enzymatic assays are merely performed, and the results included in a table.
In my opinion, although the chosen topic and the applied methods of this MS meet the interests of the readers of Molecules, in its present form the MS does not yet attain the standards of this journal. English revision is needed (grammar, typos, lack of words).
DETAILED COMMENTS
Introduction
The state of the art on triazine derivatives is not complete (e.g. Bioorg. Chem., 2018, 77, 223-235, Bioorg. Chem., 2019, 84, 363-371) and the authors should strengthen the introduction.
Line 111 – BACE1 is not a cholinesterase
In Figure 1, instead of displaying 8 structures, only one general structure should be shown, indicating the different substituents.
Results and Discussion
The rationale of the design of these compounds is not explained anywhere, namely how the different substituent groups were chosen.
Also, the discussion contained in section 2.2.1 is rather poor. In fact, a structure activity relationship (SAR) evaluation must be done. Docking studies that could elucidate the interaction inhibitor - enzyme active site would also enrich the discussion in this MS.
Conclusion
Because of the rather poor discussion part, the conclusions must be redone and transformed in a more representative section of the work performed by the authors
References
The references must be numbered.
Author Response
Thank you for your remarks.
The revised manuscript contains answers to your comments. We hope that these major revisions were made to the text will allow the work to be published quickly in Molecules.
We are (main author and co-author) a very grateful for the effort of reviewing the manuscript. The manuscript was carefully polish to make the results of this study clear to the readers. We hope that the changes introduced will be recognized as modifications that significantly improve the quality and understanding of the review.
,,In my opinion, although the chosen topic and the applied methods of this MS meet the interests of the readers of Molecules, in its present form the MS does not yet attain the standards of this journal. English revision is needed (grammar, typos, lack of words).,,
Response: English language and style were improved. Purple indicates corrections.
Introduction
,,The state of the art on triazine derivatives is not complete (e.g. Bioorg. Chem., 2018, 77, 223-235, Bioorg. Chem., 2019, 84, 363-371) and the authors should strengthen the introduction.,,
Response: The literature indicated was added. The introduction was modified.
,,Line 111 – BACE1 is not a cholinesterase,,
Response: We have correct it.
,,In Figure 1, instead of displaying 8 structures, only one general structure should be shown, indicating the different substituents.,,
Response: We have correct it.
Results and Discussion
,,The rationale of the design of these compounds is not explained anywhere, namely how the different substituent groups were chosen.,,
Response: We have correct it. We have added paragraph.
When designing the structures of compounds 4a-4h, it was assumed that in the first step the compounds will be obtained in which the nucleophilic functional groups in the side chains of amino acids will be protected in order to eliminate the possibility of a reactive chloroethylamine group with nucleophilic centers. Therefore, in the synthesized derivatives, protective groups are present on the amino, hydroxyl and carboxyl functions of amino acids. In addition, when designing the structures, the incorporation of basic amino acid residues (lysine, arginine, histidine) into the compounds was taken into account, because it was expected that these amino acids would form ionic bonds with carboxylate anions in the active center of enzymes and facilitate the interaction with phospholipid cell membranes. The incorporation of aromatic amino acid residues (derivative 4c, 4g, 4h, and 4d) with an aromatic ring on the hydroxyl group was aimed at checking the influence of π-π or hydrophobic interactions between the substrate and the active site of the enzyme. On the other hand, it was expected that it would be possible to test the influence of aromatic amino acids on the ability to inhibit β-secretase (BACE1) activity, which is known to hydrolyze beta-site amyloid precursor proteins, and the formation of β-amyloid structures results in the formation of insoluble amyloid deposits. It is known that both β-amyloid and other aggregates of their hot-spots contain aromatic amino acids [39-42].
,,Also, the discussion contained in section 2.2.1 is rather poor. In fact, a structure activity relationship (SAR) evaluation must be done,,
Response: We have correct it. The introduction was modified.
,,Docking studies that could elucidate the interaction inhibitor - enzyme active site would also enrich the discussion in this MS.,,
Response: Molecular docking is planned for future publications. This publication contains only preliminary studies
Conclusion
,,Because of the rather poor discussion part, the conclusions must be redone and transformed in a more representative section of the work performed by the authors,,
Response: We have correct it. The conclusion was modified.
,,The references must be numbered.,,
Response: We have correct it.

Reviewer 2 Report
The paper entitled “1,3,5-Triazine Nitrogen Mustards with Different Peptide Group as Innovative Candidates for AChE and BACE1 Inhibitors” by D. Drozdowska et al describes a series of trisubstituted triazines as dual AChE/BACE1 inhibitors. These compounds, which share a methoxy group and a β-chloroethylpiperazinyl fragment and display different dipeptide residues, were proposed as potential multitarget therapeutic agents for AD.
Here, the authors explore a new pharmacological application for nitrogen mustard/oligopeptide hybrids based on 1,3,5-triazine scaffold, a kind of compounds with great ability to interact with several targets and previously described as cytotoxic agents.
The topic seems to be relevant and aligned with the scope of the journal “Molecules”. However, the manuscript is not rigorous, contains many mistakes and is tedious to read. Therefore, I consider that this document is not acceptable for publication in its current form.
Comments and suggestions:
- The introduction part is neither well written nor well focused. It contains several paragraphs in which the authors describe a lot of details about the cytotoxic properties of this type of triazine analogues which lack of interest for this paper (lines 71-92). The introduction part should be modified.
- It is very difficult to check the bibliography because the numbers have not been included in the references list. Please, correct this.
- Rearrange elements and arrows in Scheme 1. Include reaction conditions and yields in the captions.
- Replace “forester resonance energy transfer” by “Föster resonance energy transfer”.
- Check the phase “and the mean percent of inhibition of AChE and BACE1 enzymatic activity were calculated as IC50 (μM)” clarifying the concept of IC50 (half maximal inhibitory concentration).
- Replace “tercine” by “tacrine” on Table 1 and line 143.
- In the discussion part the authors indicate that compound A is the most active. However, this is not entirely true since 4a has almost the same activity against AChE. This part is a little bit poor. Molecular modeling studies (docking) could contribute to explain the differences found on activity.
- In the Experimental part, check and correct the type of NMR spectrometer (250 300 or 700 MHz?).
Revise and correct all the manuscript because it contains many grammar mistakes and several typos.
Author Response
Thank you for your remarks.
The revised manuscript contains answers to your comments. We hope that these major revisions were made to the text will allow the work to be published quickly in Molecules.
We are (main author and co-author) a very grateful for the effort of reviewing the manuscript. The manuscript was carefully polish to make the results of this study clear to the readers. We hope that the changes introduced will be recognized as modifications that significantly improve the quality and understanding of the review.
- The introduction part is neither well written nor well focused. It contains several paragraphs in which the authors describe a lot of details about the cytotoxic properties of this type of triazine analogues which lack of interest for this paper (lines 71-92). The introduction part should be modified.
Response:. The introduction was modified. We have added 2 position of literature:
Bioorg. Chem., 2018, 77, 223-235, Bioorg. Chem., 2019, 84, 363-371.
In the introduction, we wanted to highlight the diversity of triazines. We presented that they are multi-target compounds and it is worth to search for further properties of these compounds.
- It is very difficult to check the bibliography because the numbers have not been included in the references list. Please, correct this.
Response: We have correct it.
- Rearrange elements and arrows in Scheme 1. Include reaction conditions and yields in the captions.
Response: We have correct it.
- Replace “forester resonance energy transfer” by “Föster resonance energy transfer”.
Response: We have correct it. We have replace “forester resonance energy transfer” by “Föster resonance energy transfer”.
- Check the phase “and the mean percent of inhibition of AChE and BACE1 enzymatic activity were calculated as IC50 (μM)” clarifying the concept of IC50 (half maximal inhibitory concentration).
Response: We have correct it.
- Replace “tercine” by “tacrine” on Table 1 and line 143.
Response: We have correct it. We have replace “tercine” by “tacrine”.
- In the discussion part the authors indicate that compound A is the most active. However, this is not entirely true since 4a has almost the same activity against AChE. This part is a little bit poor. Molecular modeling studies (docking) could contribute to explain the differences found on activity.
Response: We have correct it. Molecular docking is planned for future publications. This publication contains only preliminary studies.
- In the Experimental part, check and correct the type of NMR spectrometer (250 300 or 700 MHz?).
Response: We have correct it. Type of NMR spectrometer: 700 MHz.
Revise and correct all the manuscript because it contains many grammar mistakes and several typos.
Response: English language and style were improved. Purple indicates corrections.

Reviewer 3 Report
This manuscript describes the synthesis of 1,3,5-triazine nitrogen mustards with different peptide group as innovative candidates for AchE and BACE1 inhibitors and assesses in vitro cytotoxicity. Eight compounds were synthesized and shown to have lower cytotoxicity than the clinically evaluated compounds Donepezil, Tacrine and Quercetin. However, the inhibition of both enzymes, AchE and BACE1, is largely meaningless without a normal cell control, as the compounds may preferentially affect normal cells too. For these compounds to be suitable for Alzheimer’s diseases, they need to show specific enzyme selectivity. Therefore, the manuscript would be considerably check the effects on normal cells or even better by using an in vivo model and comparing inhibition and selectivity of these compounds.
Minor revisions:
This manuscript is worth posting on ‘Molecules’ after carefully modifying English. Also improve the quality of synthetic scheme, there is a bad graphic resolution to polish.
Author Response
Thank you for your remarks.
The revised manuscript contains answers to your comments. We hope that these minor revisions were made to the text will allow the work to be published quickly in Molecules.
We are (main author and co-author) a very grateful for the effort of reviewing the manuscript. The manuscript was carefully polish to make the results of this study clear to the readers. We hope that the changes introduced will be recognized as modifications that significantly improve the quality and understanding of the review.
This manuscript describes the synthesis of 1,3,5-triazine nitrogen mustards with different peptide group as innovative candidates for AchE and BACE1 inhibitors and assesses in vitro cytotoxicity. Eight compounds were synthesized and shown to have lower cytotoxicity than the clinically evaluated compounds Donepezil, Tacrine and Quercetin. However, the inhibition of both enzymes, AchE and BACE1, is largely meaningless without a normal cell control, as the compounds may preferentially affect normal cells too. For these compounds to be suitable for Alzheimer’s diseases, they need to show specific enzyme selectivity. Therefore, the manuscript would be considerably check the effects on normal cells or even better by using an in vivo model and comparing inhibition and selectivity of these compounds.
Response: This publication contains only preliminary studies. We are planing to develop our research in future publications. the above communication contains only synthesis and enzymatic studies. In future reports we plan to present molecular docking and also cytotoxicity experiments an in vivo model.
Minor revisions:
This manuscript is worth posting on ‘Molecules’ after carefully modifying English. Also improve the quality of synthetic scheme, there is a bad graphic resolution to polish.
Response: English language and style were improved. Purple indicates corrections.
Round 2
Reviewer 1 Report
The authors have addressed the queries of this referee and the MS has been improved. Nevertheless, in my opinion, the MS still needs minor revision to attain the standards of Molecules.
In fact, in Introduction I find quite excessive to have two big Figures with formula of compounds synthesized and studied by other authors. I advise the authors to resume the state of the art and keep maximum one figure relative to that theme.
English revision is still needed (typos, lack of words).
Author Response
Thank you for your remarks.
The revised manuscript contains answers to your comments. We hope that these minor revisions were made to the text will allow the work to be published quickly in Molecules.
We are (main author and co-author) a very grateful for the effort of reviewing the manuscript. The manuscript was carefully polish to make the results of this study clear to the readers. We hope that the changes introduced will be recognized as modifications that significantly improve the quality and understanding of the review.
Reviewer 1
The authors have addressed the queries of this referee and the MS has been improved. Nevertheless, in my opinion, the MS still needs minor revision to attain the standards of Molecules.
In fact, in Introduction I find quite excessive to have two big Figures with formula of compounds synthesized and studied by other authors. I advise the authors to resume the state of the art and keep maximum one figure relative to that theme.
We have corrected it. The introduction has been corrected. The figures have been corrected.
English revision is still needed (typos, lack of words).
We have corrected it.

Reviewer 2 Report
The paper has been checked by the authors and it has been corrected in some aspects, such as the References section and Experimental part. However, the new manuscript needs an additional revision and modification to achieve the journal quality standards.
Suggestions and comments:
- Shorten the first part of introduction in order to include in the last part the aspects about the design of target compounds, which here were described in the Chemical part, also justifying the inclusion of compound A in this study. The first part of introduction continues to contain a lot of details about the cytotoxic properties of this type of triazine analogues which lack of interest for this paper. In addition, in my opinion, the comments about 1,2,4-triazines described by Yazdani et al and Iraji et al are excessive.
- Clarify Figure 3 showing the link position of R groups in the general structure of target compounds.
- The authors incorporate in this new manuscript a simulation study of several structural parameters that could be related to drug-like properties of the titled compounds. However, they do not mention this (Lipinski rules). Why compound B is also selected for this prediction study?.
- In the Chemistry part, the expression “The incorporation of aromatic amino acid residues (derivative 4c, 4g, 4h, and 4d) with an aromatic ring on the hydroxyl group was aimed” is not clear.
- Please, include the solvents in Scheme 1.
- Check and clarify the SAR discussion. This part is still unclear and poor. The authors justify the good results obtained with compound A by the presence of 2-chloroethylamino fragment, a group also displayed by all studied compounds (4a-h). On the other hand, the activity data of compounds 4a and 4h against both enzymes are very similar. Compound 4h is neither mentioned in the abstract nor in the conclusions
- The phrase “the mean half maximal inhibitory concentration of inhibition of AChE/BACE1 enzymatic activity were calculated as IC50 (μM)” is repeated several times throughout the document, one would be enough to explain the CI50 meaning.
- Common drug names should be written in lowercase
The English language used by the authors in the manuscript must be corrected again because it contains many grammatical mistakes and typos. Please, revise and correct the following phrases and expressions: “AChE was reported to co-localize with A in neuritic plaques and can 80 enhance the rate of A fibrils formation” (could be Abeta peptide?); “of potent multitarget ligands for..” (Line 90 and 96); “of our compounds” (line 128); “ability” (Line 147); “neither compound” (Line 208); “chloroethyloamino moiety” (in several parts of the manuscript)…
Author Response
Thank you for your remarks.
The revised manuscript contains answers to your comments. We hope that these minor revisions were made to the text will allow the work to be published quickly in Molecules.
We are (main author and co-author) a very grateful for the effort of reviewing the manuscript. The manuscript was carefully polish to make the results of this study clear to the readers. We hope that the changes introduced will be recognized as modifications that significantly improve the quality and understanding of the review.
Reviewer 2
The paper has been checked by the authors and it has been corrected in some aspects, such as the References section and Experimental part. However, the new manuscript needs an additional revision and modification to achieve the journal quality standards.
Suggestions and comments:
- Shorten the first part of introduction in order to include in the last part the aspects about the design of target compounds, which here were described in the Chemical part, also justifying the inclusion of compound A in this study. The first part of introduction continues to contain a lot of details about the cytotoxic properties of this type of triazine analogues which lack of interest for this paper. In addition, in my opinion, the comments about 1,2,4-triazines described by Yazdani et al and Iraji et al are excessive.
We have corrected it.
- Clarify Figure 3 showing the link position of R groups in the general structure of target compounds.
We have corrected it.
- The authors incorporate in this new manuscript a simulation study of several structural parameters that could be related to drug-like properties of the titled compounds. However, they do not mention this (Lipinski rules). Why compound B is also selected for this prediction study?.
We have corrected it.
- In the Chemistry part, the expression “The incorporation of aromatic amino acid residues (derivative 4c, 4g, 4h, and 4d) with an aromatic ring on the hydroxyl group was aimed” is not clear.
We have corrected it.
- Please, include the solvents in Scheme 1.
We have corrected it.
- Check and clarify the SAR discussion. This part is still unclear and poor. The authors justify the good results obtained with compound A by the presence of 2-chloroethylamino fragment, a group also displayed by all studied compounds (4a-h). On the other hand, the activity data of compounds 4a and 4h against both enzymes are very similar. Compound 4h is neither mentioned in the abstract nor in the conclusions
We have corrected it.
- The phrase “the mean half maximal inhibitory concentration of inhibition of AChE/BACE1 enzymatic activity were calculated as IC50 (μM)” is repeated several times throughout the document, one would be enough to explain the CI50 meaning.
We have corrected it.
- Common drug names should be written in lowercase
We have corrected it.
The English language used by the authors in the manuscript must be corrected again because it contains many grammatical mistakes and typos. Please, revise and correct the following phrases and expressions: “AChE was reported to co-localize with A in neuritic plaques and can 80 enhance the rate of A fibrils formation” (could be Abeta peptide?); “of potent multitarget ligands for..” (Line 90 and 96); “of our compounds” (line 128); “ability” (Line 147); “neither compound” (Line 208); “chloroethyloamino moiety” (in several parts of the manuscript).
We have corrected it.
